# Blockchain for Electronic Voting System—Review and Open Research Challenges

**DOI:** 10.3390/s21175874

**Published:** 2021-08-31

**Authors:** Uzma Jafar, Mohd Juzaiddin Ab Aziz, Zarina Shukur

**Affiliations:** Faculty of Information Science and Technology, The National University of Malaysia, Bangi 43600, Malaysia; juzaiddin@ukm.edu.my (M.J.A.A.); zarinashukur@ukm.edu.my (Z.S.)

**Keywords:** electronic voting, security, blockchain-based electronic voting, privacy, blockchain technology, voting, trust

## Abstract

Online voting is a trend that is gaining momentum in modern society. It has great potential to decrease organizational costs and increase voter turnout. It eliminates the need to print ballot papers or open polling stations—voters can vote from wherever there is an Internet connection. Despite these benefits, online voting solutions are viewed with a great deal of caution because they introduce new threats. A single vulnerability can lead to large-scale manipulations of votes. Electronic voting systems must be legitimate, accurate, safe, and convenient when used for elections. Nonetheless, adoption may be limited by potential problems associated with electronic voting systems. Blockchain technology came into the ground to overcome these issues and offers decentralized nodes for electronic voting and is used to produce electronic voting systems mainly because of their end-to-end verification advantages. This technology is a beautiful replacement for traditional electronic voting solutions with distributed, non-repudiation, and security protection characteristics. The following article gives an overview of electronic voting systems based on blockchain technology. The main goal of this analysis was to examine the current status of blockchain-based voting research and online voting systems and any related difficulties to predict future developments. This study provides a conceptual description of the intended blockchain-based electronic voting application and an introduction to the fundamental structure and characteristics of the blockchain in connection to electronic voting. As a consequence of this study, it was discovered that blockchain systems may help solve some of the issues that now plague election systems. On the other hand, the most often mentioned issues in blockchain applications are privacy protection and transaction speed. For a sustainable blockchain-based electronic voting system, the security of remote participation must be viable, and for scalability, transaction speed must be addressed. Due to these concerns, it was determined that the existing frameworks need to be improved to be utilized in voting systems.

## 1. Introduction

Electoral integrity is essential not just for democratic nations but also for state voter’s trust and liability. Political voting methods are crucial in this respect. From a government standpoint, electronic voting technologies can boost voter participation and confidence and rekindle interest in the voting system. As an effective means of making democratic decisions, elections have long been a social concern. As the number of votes cast in real life increases, citizens are becoming more aware of the significance of the electoral system [1,2]. The voting system is the method through which judges judge who will represent in political and corporate governance. Democracy is a system of voters to elect representatives by voting [3,4]. The efficacy of such a procedure is determined mainly by the level of faith that people have in the election process. The creation of legislative institutions to represent the desire of the people is a well-known tendency. Such political bodies differ from student unions to constituencies. Over the years, the vote has become the primary resource to express the will of the citizens by selecting from the choices they made [2].

The traditional or paper-based polling method served to increase people’s confidence in the selection by majority voting. It has helped make the democratic process and the electoral system worthwhile for electing constituencies and governments more democratized. There are 167 nations with democracy in 2018, out of approximately 200, which are either wholly flawed or hybrid [5,6]. The secret voting model has been used to enhance trust in democratic systems since the beginning of the voting system.

It is essential to ensure that assurance in voting does not diminish. A recent study revealed that the traditional voting process was not wholly hygienic, posing several questions, including fairness, equality, and people’s will, was not adequately [7] quantified and understood in the form of government [2,8].

Engineers across the globe have created new voting techniques that offer some anti-corruption protection while still ensuring that the voting process should be correct. Technology introduced the new electronic voting techniques and methods [9], which are essential and have posed significant challenges to the democratic system. Electronic voting increases election reliability when compared to manual polling. In contrast to the conventional voting method, it has enhanced both the efficiency and the integrity of the process [10]. Because of its flexibility, simplicity of use, and cheap cost compared to general elections, electronic voting is widely utilized in various decisions [11]. Despite this, existing electronic voting methods run the danger of over-authority and manipulated details, limiting fundamental fairness, privacy, secrecy, anonymity, and transparency in the voting process. Most procedures are now centralized, licensed by the critical authority, controlled, measured, and monitored in an electronic voting system, which is a problem for a transparent voting process in and of itself.

On the other hand, the electronic voting protocols have a single controller that oversees the whole voting process [12]. This technique leads to erroneous selections due to the central authority’s dishonesty (election commission), which is difficult to rectify using existing methods. The decentralized network may be used as a modern electronic voting technique to circumvent the central authority.

Blockchain technology offers a decentralized node for online voting or electronic voting. Recently distributed ledger technologies such blockchain were used to produce electronic voting systems mainly because of their end-to-end verification advantages [13]. Blockchain is an appealing alternative to conventional electronic voting systems with features such as decentralization, non-repudiation, and security protection. It is used to hold both boardroom and public voting [8]. A blockchain, initially a chain of blocks, is a growing list of blocks combined with cryptographic connections. Each block contains a hash, timestamp, and transaction data from the previous block. The blockchain was created to be data-resistant. Voting is a new phase of blockchain technology; in this area, the researchers are trying to leverage benefits such as transparency, secrecy, and non-repudiation that are essential for voting applications [14]. With the usage of blockchain for electronic voting applications, efforts such as utilizing blockchain technology to secure and rectify elections have recently received much attention [15].

The remainder of the paper is organized as follows. Section 2 explains how blockchain technology works, and a complete background of this technology is discussed. How blockchain technology can transfer the electronic voting system is covered in Section 3. In Section 4, the problems and their solutions of developing online voting systems are identified. The security requirements for the electronic voting system are discussed in Section 5, and the possibility of electronic voting on blockchain is detailed in Section 6. Section 7 discusses the available blockchain-based electronic voting systems and analyzes them thoroughly. In Section 8, all information related to the latest literature review is discussed and analyzed deeply. Section 9 addresses the study, open issues, and future trends. Furthermore, in the end, Section 10 concludes this survey.

## 2. Background

The first things that come to mind about the blockchain are cryptocurrencies and smart contracts because of the well-known initiatives in Bitcoin and Ethereum. Bitcoin was the first crypto-currency solution that used a blockchain data structure. Ethereum introduced smart contracts that leverage the power of blockchain immutability and distributed consensus while offering a crypto-currency solution comparable to Bitcoin. The concept of smart contracts was introduced much earlier by Nick Szabo in the 1990s and is described as “a set of promises, specified in digital form, including protocols within which the parties perform on these promises” [16]. In Ethereum, a smart contract is a piece of code deployed to the network so that everyone has access to it. The result of executing this code is verified by a consensus mechanism and by every member of the network as a whole [17].

Today, we call a blockchain a set of technologies combining the blockchain data structure itself, distributed consensus algorithm, public key cryptography, and smart contracts [18]. Below we describe these technologies in more detail.

Blockchain creates a series of blocks replicated on a peer-to-peer network. Any block in blockchain has a cryptographic hash and timestamp added to the previous block, as shown in Figure 1. A block contains the Merkle tree block header and several transactions [19]. It is a secure networking method that combines computer science and mathematics to hide data and information from others that is called cryptography. It allows the data to be transmitted securely across the insecure network, in encrypted and decrypted forms [20,21].

As was already mentioned, the blockchain itself is the name for the data structure. All the written data are divided into blocks, and each block contains a hash of all the data from the previous block as part of its data [22]. The aim of using such a data structure is to achieve provable immutability. If a piece of data is changed, the block’s hash containing this piece needs to be recalculated, and the hashes of all subsequent blocks also need to be recalculated [23]. It means only the hash of the latest block has to be used to guarantee that all the data remains unchanged. In blockchain solutions, data stored in blocks are formed from all the validated transactions during their creation, which means no one can insert, delete or alter transactions in an already validated block without it being noticed [24]. The initial zero-block, called the “genesis block,” usually contains some network settings, for example, the initial set of validators (those who issue blocks).

Blockchain solutions are developed to be used in a distributed environment. It is assumed that nodes contain identical data and form a peer-to-peer network without a central authority. A consensus algorithm is used to reach an agreement on blockchain data that is fault-tolerant in the presence of malicious actors. Such consensus is called Byzantine fault tolerance, named after the Byzantine Generals’ Problem [25]. Blockchain solutions use different Byzantine fault tolerance (BFT) consensus algorithms: Those that are intended to be used in fully decentralized self-organizing networks, such as cryptocurrency platforms, use algorithms such as proof-of-work or proof-of-stake, where validators are chosen by an algorithm so that it is economically profitable for them to act honestly [26]. When the network does not need to be self-organized, validators can be chosen at the network setup stage [27]. The point is that all validators execute all incoming transactions and agree on achieving results so that more than two-thirds of honest validators need to decide on the outcome.

Public key cryptography is used mainly for two purposes: Firstly, all validators own their keypairs used to sign consensus messages, and, secondly, all incoming transactions (requests to modify blockchain data) have to be signed to determine the requester. Anonymity in a blockchain context relates to the fact that anyone wanting to use cryptocurrencies just needs to generate a random keypair and use it to control a wallet linked to a public key [28]. The blockchain solution guarantees that only the keypair owner can manage the funds in the wallet, and this property is verifiable [29,30]. As for online voting, ballots need to be accepted anonymously but only from eligible voters, so a blockchain by itself definitely cannot solve the issue of voter privacy.

Smart contracts breathed new life into blockchain solutions. They stimulated the application of blockchain technology in efforts to improve numerous spheres. A smart contract itself is nothing more than a piece of logic written in code. Still, it can act as an unconditionally trusted third party in conjunction with the immutability provided by a blockchain data structure and distributed consensus [31]. Once written, it cannot be altered, and all the network participants verify all steps. The great thing about smart contracts is that anybody who can set up a blockchain node can verify its outcome. 

As is the case with any other technology, blockchain technology has its drawbacks. Unlike other distributed solutions, a blockchain is hard to scale: An increasing number of nodes does not improve network performance because, by definition, every node needs to execute all transactions, and this process is not shared among the nodes [32]. Moreover, increasing the number of validators impacts performance because it implies a more intensive exchange of messages during consensus. For the same reason, blockchain solutions are vulnerable to various denial-of-service attacks. If a blockchain allows anyone to publish smart contracts in a network, then the operation of the entire network can be disabled by simply putting an infinite loop in a smart contract. A network can also be attacked by merely sending a considerable number of transactions: At some point, the system will refuse to receive anything else. In cryptocurrency solutions, all transactions have an execution cost: the more resources a transaction utilizes, the more expensive it will be, and there is a cost threshold, with transactions exceeding the threshold being discarded. In private blockchain networks [33,34], this problem is solved depending on how the network is implemented via the exact mechanism of transaction cost, access control, or something more suited to the specific context.

### 2.1. Core Components of Blockchain Architecture

These are the main architectural components of Blockchain as shown in Figure 2.

Node: Users or computers in blockchain layout (every device has a different copy of a complete ledger from the blockchain);Transaction: It is the blockchain system’s smallest building block (records and details), which blockchain uses;Block: A block is a collection of data structures used to process transactions over the network distributed to all nodes.Chain: A series of blocks in a particular order;Miners: Correspondent nodes to validate the transaction and add that block into the blockchain system;Consensus: A collection of commands and organizations to carry out blockchain processes.

### 2.2. Critical Characteristics of Blockchain Architecture

Blockchain architecture has many benefits for all sectors that incorporate blockchain. Here are a variety of embedded characteristics as described Figure 3:Cryptography: Blockchain transactions are authenticated and accurate because of computations and cryptographic evidence between the parties involved;Immutability: Any blockchain documents cannot be changed or deleted;Provenance: It refers to the fact that every transaction can be tracked in the blockchain ledger;Decentralization: The entire distributed database may be accessible by all members of the blockchain network. A consensus algorithm allows control of the system, as shown in the core process;Anonymity: A blockchain network participant has generated an address rather than a user identification. It maintains anonymity, especially in a blockchain public system;Transparency: It means being unable to manipulate the blockchain network. It does not happen as it takes immense computational resources to erase the blockchain network.

## 3. How Blockchain Can Transform the Electronic Voting System

Blockchain technology fixed shortcomings in today’s method in elections made the polling mechanism clear and accessible, stopped illegal voting, strengthened the data protection, and checked the outcome of the polling. The implementation of the electronic voting method in blockchain is very significant [35]. However, electronic voting carries significant risks such as if an electronic voting system is compromised, all cast votes can probably be manipulated and misused. Electronic voting has thus not yet been adopted on a national scale, considering all its possible advantages. Today, there is a viable solution to overcome the risks and electronic voting, which is blockchain technology. In Figure 4, one can see the main difference between both of the systems. In traditional voting systems, we have a central authority to cast a vote. If someone wants to modify or change the record, they can do it quickly; no one knows how to verify that record. One does not have the central authority; the data are stored in multiple nodes. It is not possible to hack all nodes and change the data. Thus, in this way, one cannot destroy the votes and efficiently verify the votes by tally with other nodes.

If the technology is used correctly, the blockchain is a digital, decentralized, encrypted, transparent ledger that can withstand manipulation and fraud. Because of the distributed structure of the blockchain, a Bitcoin electronic voting system reduces the risks involved with electronic voting and allows for a tamper-proof for the voting system. A blockchain-based electronic voting system requires a wholly distributed voting infrastructure. Electronic voting based on blockchain will only work where the online voting system is fully controlled by no single body, not even the government [36]. To sum-up, elections can only be free and fair when there is a broad belief in the legitimacy of the power held by those in positions of authority. The literature review for this field of study and other related experiments may be seen as a good path for making voting more efficient in terms of administration and participation. However, the idea of using blockchain offered a new model for electronic voting.

## 4. Problems and Solutions of Developing Online Voting Systems

Whether talking about traditional paper-based voting, voting via digital voting machines, or an online voting system, several conditions need to be satisfied:Eligibility: Only legitimate voters should be able to take part in voting;Unreusability: Each voter can vote only once;Privacy: No one except the voter can obtain information about the voter’s choice;Fairness: No one can obtain intermediate voting results;Soundness: Invalid ballots should be detected and not taken into account during tallying;Completeness: All valid ballots should be tallied correctly.

Below is a brief overview of the solutions for satisfying these properties in online voting systems.

### 4.1. Eligibility

The solution to the issue of eligibility is rather apparent. To take part in online voting, voters need to identify themselves using a recognized identification system. The identifiers of all legitimate voters need to be added to the list of participants. But there are threats: Firstly, all modifications made to the participation list need to be checked so that no illegitimate voters can be added, and secondly, the identification system should be both trusted and secure so that a voter’s account cannot be stolen or used by an intruder. Building such an identification system is a complex task in itself [37]. However, because this sort of system is necessary for a wide range of other contexts, especially related to digital government services, researchers believe it is best to use an existing identification system, and the question of creating one is beyond the scope of work.

### 4.2. Unreusability

At first, glance, implementing unreusability may seem straightforward—when a voter casts their vote, all that needs to be done is to place a mark in the participation list and not allow them to vote a second time. But privacy needs to be taken into consideration; thus, providing both unreusability and voter anonymity is tricky. Moreover, it may be necessary to allow the voter to re-vote, making the task even more complex [38]. A brief overview of unreusability techniques will be provided below in conjunction with the outline on implementing privacy.

### 4.3. Privacy

Privacy in the context of online voting means that no one except the voter knows how a participant has voted. Achieving this property mainly relies on one (or more) of the following techniques: blind signatures, homomorphic encryption, and mix-networks [39]. Blind signature is a method of signing data when the signer does not know what they are signing. It is achieved by using a blinding function so that blinding and signing functions are commutative–Blind(Sign(message)) = Sign(Blind(message)). The requester blinds (applies blinding function to) their message and sends it for signing. After obtaining a signature for a blinded message, they use their knowledge of blinding parameters to derive a signature for an unblinded message. Blind signatures mathematically prevent anyone except the requester from linking a blinded message and a corresponding signature pair with an unblinded one [40].

The voting scheme proposed by Fujioka, Okamoto, and Ohta in 1992 [41] uses a blind signature: An eligible voter blinds his ballot and sends it to the validator. The validator verifies that the voter is allowed to participate, signs the blinded ballot, and returns it to the voter. The voter then derives a signature for the unblinded vote and sends it to the tallier, and the tallier verifies the validator’s signature before accepting the ballot. 

Many online voting protocols have evolved from this scheme, improving usability (in the original method, the voter had to wait till the end of the election and send a ballot decryption key), allowing re-voting, or implementing coercion resistance. The main threat here is the power of the signer: There must be a verifiable log of all emitted signatures; this information logically corresponds to the receiving of a ballot by the voter, so it should be verified that only eligible voters receive signatures from the signer [42]. It should also be verifiable that accounts of voters who are permitted to vote but have not taken part in voting are not utilized by an intruder. To truly break the link between voter and ballot, the ballot and the signature need to be sent through an anonymous channel [43]. 

Homomorphic encryption is a form of encryption that allows mathematical operations to be performed on encrypted data without decryption, for example, the addition

Enc(a) + Enc(b) = Enc(a + b); or multiplication Enc(a) × Enc(b) = Enc(a × b). In the context of online voting, additive homomorphic encryption allows us to calculate the sum of all the voters’ choices before decryption. 

It is worth mentioning here that multiplicative homomorphic encryption can generally be used as an additive. For example, if we have choices x and y and multiplicative homomorphic encryption, we can select a value g and encrypt exponentiation: Enc(gx) × Enc(gy) = Enc(g(x + y)).

Homomorphic encryption can be used to obtain various properties necessary in an online voting system; with regards to privacy, it is used so that only the sum of all the choices is decrypted, and never each voter’s choice by itself. Using homomorphic encryption for privacy implies that decryption is performed by several authorities so that no one can obtain the decryption key; otherwise, privacy will be violated [44]. 

It is usually implemented with a threshold decryption scheme. For instance, let us say that we have *n* authorities. To decrypt a result, we need t of them, t <= *n*. The protocol assumes that each authority applies its vital part to the sum of the encrypted choices. After t authorities perform this operation, we get the decrypted total sum of choices. In contrast to the blind signature scheme, no anonymous channel between voters and the system is needed. Still, privacy relies on trust in the authorities: If a malicious agreement is reached, all voters can be deanonymized. 

Mix-networks also rely on the distribution of the trust, but in another way. The idea behind a mix-network is that voters’ choices go through several mix-servers that shuffle them and perform an action–either decryption or re-encryption, depending on the mix-network type. In a decryption mix network, each mixing server has its key, and the voter encrypts their choice like an onion so that each server will unwrap its layer of decryption. In re-encryption mix-networks, each mix server re-encrypts the voters’ choices. 

There are many mix-network proposals, and reviewing all their properties is beyond the scope of this paper. The main point regarding privacy here is that, in theory, if at least one mix-server performs an honest shuffle, privacy is preserved. It is slightly different from privacy based on homomorphic encryption, where we make assumptions about the number of malicious authorities. In addition, the idea behind mix-networks can be used to build anonymous channels required by other techniques [45].

### 4.4. Fairness

Fairness in terms of no one obtaining intermediate results is achieved straightforwardly: Voters encrypt their choices before sending, and those choices are decrypted at the end of the voting process. The critical thing to remember here is that if someone owns a decryption key with access to encrypted decisions, they can obtain intermediate results. This problem is solved by distributing the key among several keyholders [41]. A system where all the key holders are required for decryption is unreliable—if one of the key holders does not participate, decryption cannot be performed. Therefore, threshold schemes are used whereby a specific number of key holders are required to perform decryption. There are two main approaches for distributing the key: secret sharing, where a trusted dealer divides the generated key into parts and distributes them among key holders (e.g., Shamir’s Secret Sharing protocol); and distributed key generation, where no trusted dealer is needed, and all parties contribute to the calculation of the key (for example, Pedersen’s Distributed Key Generation protocol).

### 4.5. Soundness and Completeness

On the face of it, the completeness and soundness properties seem relatively straightforward, but realizing them can be problematic depending on the protocol. If ballots are decrypted one by one, it is easy to distinguish between valid and invalid ones, but things become more complicated when it comes to homomorphic encryption. As a single ballot is never decrypted, the decryption result will not show if more than one option was chosen or if the poll was formed so that it was treated as ten choices (or a million) at once. Thus, we need to prove that the encrypted data meets the properties of a valid ballot without compromising any information that can help determine how the vote was cast. This task is solved by zero-knowledge proof [46]. By definition, this is a cryptographic method of proving a statement about the value without disclosing the value itself. More specifically, range proofs demonstrate that a specific value belongs to a particular set in such cases. 

The properties described above are the bare minimum for any voting solution. But all the technologies mentioned above are useless if there is no trust in the system itself. A voting system needs to be fully verifiable to earn this trust, i.e., everyone involved can ensure that the system complies with the stated properties. Ensuring verifiability can be split into two tasks: personal, when the voter can verify that their ballot is correctly recorded and tallied; and universal, when everyone can prove that the system as a whole works precisely [47]. This entails the inputs and outputs of the voting protocol stages being published and proof of correct execution. For example, mix-networks rely on proof of correct shuffling (a type of zero-knowledge proof), while proof of correct decryption is also used in mix-networks and threshold decryption. The more processes that are open to public scrutiny, the more verifiable the system is. However, online voting makes extensive use of cryptography, and the more complex the cryptography, the more obscure it is for most system users [48]. It may take a considerable amount of time to study the protocol and even more to identify any vulnerabilities or backdoors, and even if the entire system is carefully researched, there is no guarantee that the same code is used in real-time. 

Last but not least are problems associated with coercion and vote-buying. Online voting brings these problems to the next level: As ballots are cast remotely from an uncontrolled environment, coercers and vote buyers can operate on a large scale [49]. That is why one of the desired properties of an online voting system is coercion resistance. It is called resistance because nothing can stop the coercer from standing behind the voter and controlling its actions. The point here is to do as much as possible to lower the risk of mass interference. Both kinds of malefactors—coercers and vote buyers—demand proof of how a voter voted. That is why many types of coercion resistance voting schemes introduce the concept of receipt-freeness.

The voter cannot create a receipt that proves how they voted. The approaches to implementing receipt-freeness generally rely on a trusted party—either a system or device that hides the unique parameters used to form a ballot from the voter, so the voter cannot prove that a particular ballot belongs to them [50]. The reverse side of this approach is that if a voter claims that their vote is recorded or tallied incorrectly, they simply cannot prove it due to a lack of evidence. 

An overview of technologies used to meet the necessary properties of online voting systems and analysis deliberately considered the properties separately [51]. When it comes to assembling the whole protocol, most solutions introduce a trade-off. For example, as noted for the blind signature, there is a risk that non-eligible voters will vote, receipt-freeness contradicts verifiability, a more complex protocol can dramatically reduce usability, etc. Furthermore, the fundamental principles of developing the solution, but many additional aspects must be considered in a real-world system like security and reliability of the communication protocols, system deployment procedure, access to system components [52]. At present, no protocol satisfies all the desired properties and, therefore, no 100% truly robust online voting system exists.

## 5. Security Requirements for Voting System

Suitable electronic voting systems should meet the following electronic voting requirements. Figure 5 shows the main security requirements for electronic voting systems.

### 5.1. Anonymity

Throughout the polling process, the voting turnout must be secured from external interpretation. Any correlation between registered votes and voter identities inside the electoral structure shall be unknown [20,53].

### 5.2. Auditability and Accuracy

Accuracy, also called correctness, demands that the declared results correspond precisely to the election results. It means that nobody can change the voting of other citizens, that the final tally includes all legitimate votes [54], and that there is no definitive tally of invalid ballots.

### 5.3. Democracy/Singularity

A “democratic” system is defined if only eligible voters can vote, and only a single vote can be cast for each registered voter [55]. Another function is that no one else should be able to duplicate the vote.

### 5.4. Vote Privacy

After the vote is cast, no one should be in a position to attach the identity of a voter with its vote. Computer secrecy is a fragile type of confidentiality, which means that the voting relationship remains hidden for an extended period as long as the current rate continues to change with computer power and new techniques [56,57].

### 5.5. Robustness and Integrity

This condition means that a reasonably large group of electors or representatives cannot disrupt the election. It ensures that registered voters will abstain without problems or encourage others to cast their legitimate votes for themselves. The corruption of citizens and officials is prohibited from denying an election result by arguing that some other member has not performed their portion correctly [58].

### 5.6. Lack of Evidence

While anonymous privacy ensures electoral fraud safeguards, no method can be assured that votes are placed under bribery or election rigging in any way. This question has its root from the start [59].

### 5.7. Transparency and Fairness

It means that before the count is released, no one can find out the details. It avoids acts such as manipulating late voters’ decisions by issuing a prediction or offering a significant yet unfair benefit to certain persons or groups as to be the first to know [60].

### 5.8. Availability and Mobility

During the voting period, voting systems should always be available. Voting systems should not limit the place of the vote.

### 5.9. Verifiable Participation/Authenticity

The criterion also referred to as desirability [61] makes it possible to assess whether or not a single voter engaged in the election [62]. This condition must be fulfilled where voting by voters becomes compulsory under the constitution (as is the case in some countries such as Australia, Germany, Greece) or in a social context, where abstention is deemed to be a disrespectful gesture (such as the small and medium-sized elections for a delegated corporate board).

### 5.10. Accessibility and Reassurance

To ensure that everyone who wants to vote has the opportunity to avail the correct polling station and that polling station must be open and accessible for the voter. Only qualified voters should be allowed to vote, and all ballots must be accurately tallied to guarantee that elections are genuine [63].

### 5.11. Recoverability and Identification

Voting systems can track and restore voting information to prevent errors, delays, and attacks.

### 5.12. Voters Verifiability

Verifiability means that processes exist for election auditing to ensure that it is done correctly. Three separate segments are possible for this purpose: (a) uniform verification or public verification [64] that implies that anybody such as voters, governments, and external auditors can test the election after the declaration of the tally; (b) transparent verifiability against a poll [65], which is a weaker prerequisite for each voter to verify whether their vote has been taken into account properly.

## 6. Electronic Voting on Blockchain

This section provides some background information on electronic voting methods. Electronic voting is a voting technique in which votes are recorded or counted using electronic equipment. Electronic voting is usually defined as voting that is supported by some electronic hardware and software. Such regularities should be competent in supporting/implementing various functions, ranging from election setup through vote storage. Kiosks at election offices, laptops, and, more recently, mobile devices are all examples of system types. Voter registration, authentication, voting, and tallying must be incorporated in the electronic voting systems Figure 6.

One of the areas where blockchain may have a significant impact is electronic voting. The level of risk is so great that electronic voting alone is not a viable option. If an electronic voting system is hacked, the consequences will be far-reaching. Because a blockchain network is entire, centralized, open, and consensus-driven, the design of a blockchain-based network guarantees that fraud is not theoretically possible until adequately implemented [66]. As a result, the blockchain’s unique characteristics must be taken into account. There is nothing inherent about blockchain technology that prevents it from being used to any other kind of cryptocurrency. The idea of utilizing blockchain technology to create a tamper-resistant electronic/online voting network is gaining momentum [67]. End users would not notice a significant difference between a blockchain-based voting system and a traditional electronic voting system.

On the other hand, voting on the blockchain will be an encrypted piece of data that is fully open and publicly stored on a distributed blockchain network rather than a single server. A consensus process on a blockchain mechanism validates each encrypted vote, and the public records each vote on distributed copies of the blockchain ledger [68]. The government will observe how votes were cast and recorded, but this information will not be restricted to policy. The blockchain voting system is decentralized and completely open, yet it ensures that voters are protected. This implies that anybody may count the votes with blockchain electronic voting, but no one knows who voted to whom. Standard electronic voting and blockchain-based electronic voting apply to categorically distinct organizational ideas.

## 7. Current Blockchain-Based Electronic Voting Systems

The following businesses and organizations, founded but mainly formed over the last five years, are developing the voting sector. All share a strong vision for the blockchain network to put transparency into practice. Table 1 shows the different online platforms, their consensus, and the technology used to develop the system. Currently available blockchain-based voting systems have scalability issues. These systems can be used on a small scale. Still, their systems are not efficient for the national level to handle millions of transactions because they use current blockchain frameworks such as Bitcoin, Ethereum, Hyperledger Fabric, etc. In Table 2 we present scalability analysis of famous blockchain platforms. The scalability issue arises with blockchain value suggestions; therefore, altering blockchain settings cannot be easily increased. To scale a blockchain, it is insufficient to increase the block size or lower the block time by lowering the hash complexity. By each approach, the scaling capability hits a limit before it can achieve the transactions needed to compete with companies such as Visa, which manages an average of 150 million transactions per day. Research released by Tata Communications in 2018 has shown that 44% of the companies used blockchain in their survey and refers to general issues arising from the use of new technology. The unresolved scalability issue emerges as a barrier from an architectural standpoint to blockchain adoption and practical implementations. As Deloitte Insights puts it, “blockchain-based systems are comparatively slow. Blockchain’s sluggish transaction speed is a major concern for enterprises that depend on high-performance legacy transaction processing systems.” In 2017 and 2018, the public attained an idea of issues with scalability: significant delays and excessive charging for the Bitcoin network and the infamous Cryptokitties application that clogged the Ethereum blockchain network (a network that thousands of decentralized applications rely on).

### 7.1. Follow My Vote

It is a company that has a secure online voting platform cantered on the blockchain with polling box audit ability to see real-time democratic development [69]. This platform enables the voters to cast their votes remotely and safely and vote for their ideal candidate. It can then use their identification to open the ballot box literally and locate their ballot and check that both that it is correct and that the election results have been proven to be accurate mathematically.

### 7.2. Voatz

This company established a smartphone-based voting system on blockchain to vote remotely and anonymously and verify that the vote was counted correctly [70]. Voters confirm their applicants and themselves on the application and give proof by an image and their identification to include biometric confirmation that either a distinctive signature such as fingerprints or retinal scans.

### 7.3. Polyas

It was founded in Finland in 1996. The company employs blockchain technology to provide the public and private sectors with an electronic voting system [71]. Polyas has been accredited as secure enough by the German Federal Office for Information Security for electronic voting applications in 2016. Many significant companies throughout Germany use Polyas to perform electronic voting systems. Polyas now has customers throughout the United States and Europe.

### 7.4. Luxoft

The first customized blockchain electronic voting system used by a significant industry was developed by the global I.T. service provider Luxoft Harding, Inc., in partnership with the City of Zug and Lucerne University of Applied Sciences of Switzerland [72]. To drive government adoption of blockchain-based services, Luxoft announces its commitment to open source this platform and establishes a Government Alliance Blockchain to promote blockchain use in public institutions.

### 7.5. Polys

Polys is a blockchain-based online voting platform and backed with transparent crypto algorithms. Kaspersky Lab powers them. Polys supports the organization of polls by student councils, unions, and associations and helps them spread electoral information to the students [73]. Online elections with Polys lead to productivity in a community, improve contact with group leaders, and attract new supporters [74]. Polys aims to reduce time and money for local authorities, state governments, and other organizations by helping them to focus on collecting and preparing proposals.

### 7.6. Agora

It is a group that has introduced a blockchain digital voting platform. It was established in 2015 and partially implemented in the presidential election in Sierra Leone in March 2018. Agora’s architecture is built on several technological innovations: a custom blockchain, unique participatory security, and a legitimate consensus mechanism [75]. The vote is the native token in Agora’s ecosystem. It encourages citizens and chosen bodies, serving as writers of elections worldwide to commit to a secure and transparent electoral process. The vote is the Agora ecosystem’s universal token.

## 8. Related Literature Review

Several articles have been published in the recent era that highlighted the security and privacy issues of blockchain-based electronic voting systems. Reflects the comparison of selected electronic voting schemes based on blockchain.

The open vote network (OVN) was presented by [76], which is the first deployment of a transparent and self-tallying internet voting protocol with total user privacy by using Ethereum. In OVN, the voting size was limited to 50–60 electors by the framework. The OVN is unable to stop fraudulent miners from corrupting the system. A fraudulent voter may also circumvent the voting process by sending an invalid vote. The protocol does nothing to guarantee the resistance to violence, and the electoral administrator wants to trust [77,78].

Furthermore, since solidity does not support elliptic curve cryptography, they used an external library to do the computation [79]. After the library was added, the voting contract became too big to be stored on the blockchain. Since it has occurred throughout the history of the Bitcoin network, OVN is susceptible to a denial-of-service attack [80]. Table 3 shows the main comparison of selected electronic voting schemes based on blockchain.

Lai et al. [81] suggested a decentralized anonymous transparent electronic voting system (DATE) requiring a minimal degree of confidence between participants. They think that for large-scale electronic elections, the current DATE voting method is appropriate. Unfortunately, their proposed system is not strong enough to secure from DoS attacks because there was no third-party authority on the scheme responsible for auditing the vote after the election process. This system is suitable only for small scales because of the limitation of the platform [8]. Although using Ring Signature keeps the privacy of individual voters, it is hard to manage and coordinate several signer entities. They also use PoW consensus, which has significant drawbacks such as energy consumption: the “supercomputers” of miners monitor a million computations a second, which is happening worldwide. Because this arrangement requires high computational power, it is expensive and energy-consuming.

Shahzad et al. [2] proposed the BSJC proof of completeness as a reliable electronic voting method. They used a process model to describe the whole system’s structure. On a smaller scale, it also attempted to address anonymity, privacy, and security problems in the election. However, many additional problems have been highlighted. The proof of labor, for example, is a mathematically vast and challenging job that requires a tremendous amount of energy to complete. Another problem is the participation of a third party since there is a significant risk of data tampering, leakage, and unfair tabulated results, all of which may impact end-to-end verification. On a large scale, generating and sealing the block may cause the polling process to be delayed [8].

Gao et al. [8] has suggested a blockchain-based anti-quantum electronic voting protocol with an audit function. They have also made modifications to the code-based Niederreiter algorithm to make it more resistant to quantum assaults. The Key Generation Center (KGC) is a certificateless cryptosystem that serves as a regulator. It not only recognizes the voter’s anonymity but also facilitates the audit’s functioning. However, an examination of their system reveals that, even if the number of voters is modest, the security and efficiency benefits are substantial for a small-scale election. If the number is high, some of the efficiency is reduced to provide better security [82].

Yi [83] presented the blockchain-based electronic voting Scheme (BES) that offered methods for improving electronic voting security in the peer-to-peer network using blockchain technology. A BES is based on the distributed ledger (DLT) may be employed to avoid vote falsification. The system was tested and designed on Linux systems in a P2P network. In this technique, counter-measurement assaults constitute a significant issue. This method necessitates the involvement of responsible third parties and is not well suited to centralized usage in a system with many agents. A distributed process, i.e., the utilization of secure multipart computers, may address the problem. However, in this situation, computing expenses are more significant and maybe prohibitive if the calculation function is complex and there are too many participants. [84,85].

Khan, K.M. [86] has proposed block-based e-voting architecture (BEA) that conducted strict experimentation with permissioned and permissionless blockchain architectures through different scenarios involving voting population, block size, block generation rate, and block transaction speed. Their experiments also uncovered fascinating findings of how these parameters influence the overall scalability and reliability of the electronic voting model, including interchanges between different parameters and protection and performance measures inside the organization alone. In their scheme, the electoral process requires the generation of voter addresses and candidate addresses. These addresses are then used to cast votes from voters to candidates. The mining group updates the ledger of the main blockchain to keep track of votes cast and the status of the vote. The voting status remains unconfirmed until a miner updates the main ledger. The vote is then cast using the voting machine at the polling station.

However, in this model, there are some flaws found. There is no regulatory authority to restrict invalid voters from casting a vote, and it is not secure from quantum attach. Their model is not accurate and did not care about voter’s integrity. Moreover, their scheme using Distributed consensus in which testimonies (data and facts) can be organized into cartels because fewer people keep the network active, a “51%” attack becomes easier to organize. This attack is potentially more concentrated and did not discuss scalability and delays in electronic voting, which are the main concerns about the blockchain voting system. They have used the Multichain framework, a private blockchain derived from Bitcoin, which is unsuitable for the nationwide voting process. As the authors mentioned, their system is efficient for small and medium-sized voting environments only.

## 9. Discussion and Future Work

Many issues with electronic voting can be solved using blockchain technology, which makes electronic voting more cost-effective, pleasant, and safe than any other network. Over time, research has highlighted specific problems, such as the need for further work on blockchain-based electronic voting and that blockchain-based electronic voting schemes have significant technical challenges.

### 9.1. Scalability and Processing Overheads

For a small number of users, blockchain works well. However, when the network is utilized for large-scale elections, the number of users increases, resulting in a higher cost and time consumption for consuming the transaction. Scalability problems are exacerbated by the growing number of nodes in the blockchain network. In the election situation, the system’s scalability is already a significant issue [87]. An electronic voting integration will further impact the system’s scalability based on blockchain [88,89]. Table 3 elucidates different metrics or properties inherent to all blockchain frameworks and presents a comparative analysis of some blockchain-based platforms such as Bitcoin, Ethereum, Hyperledger Fabric, Litecoin, Ripple, Dogecoin, Peercoin, etc. One way to enhance blockchain scaling would be to parallelize them, which is called sharding. In a conventional blockchain network, transactions and blocks are verified by all the participating nodes. In order to enable high concurrency in data, the data should be horizontally partitioned into parts, each known as a shard.

### 9.2. User Identity

As a username, blockchain utilizes pseudonyms. This strategy does not provide complete privacy and secrecy. Because the transactions are public, the user’s identity may be discovered by examining and analyzing them. The blockchain’s functionality is not well suited to national elections [90].

### 9.3. Transactional Privacy

In blockchain technology, transactional anonymity and privacy are difficult to accomplish [91]. However, transactional secrecy and anonymity are required in an election system due to the presence of the transactions involved. For this purpose, a third-party authority required but not centralized, this third-party authority should check and balance on privacy.

### 9.4. Energy Efficiency

Blockchain incorporates energy-intensive processes such as protocols, consensus, peer-to-peer communication, and asymmetrical encryption. Appropriate energy-efficient consensus methods are a need for blockchain-based electronic voting. Researchers suggested modifications to current peer-to-peer protocols to make them more energy-efficient [92,93].

### 9.5. Immatureness

Blockchain is a revolutionary technology that symbolizes a complete shift to a decentralized network. It has the potential to revolutionize businesses in terms of strategy, structure, processes, and culture. The current implementation of blockchain is not without flaws. The technology is presently useless, and there is little public or professional understanding about it, making it impossible to evaluate its future potential. All present technical issues in blockchain adoption are usually caused by the technology’s immaturity [94].

### 9.6. Acceptableness

While blockchain excels at delivering accuracy and security, people’s confidence and trust are critical components of effective blockchain electronic voting [95]. The intricacy of blockchain may make it difficult for people to accept blockchain-based electronic voting, and it can be a significant barrier to ultimately adopting blockchain-based electronic voting in general public acceptance [96]. A big marketing campaign needed for this purpose to provide awareness to people about the benefits of blockchain voting systems, so that it will be easy for them to accept this new technology.

### 9.7. Political Leaders’ Resistance

Central authorities, such as election authorities and government agencies, will be shifted away from electronic voting based on blockchain. As a result, political leaders who have profited from the existing election process are likely to oppose the technology because blockchain will empower social resistance through decentralized autonomous organizations [97].

## 10. Conclusions

The goal of this research is to analyze and evaluate current research on blockchain-based electronic voting systems. The article discusses recent electronic voting research using blockchain technology. The blockchain concept and its uses are presented first, followed by existing electronic voting systems. Then, a set of deficiencies in existing electronic voting systems are identified and addressed. The blockchain’s potential is fundamental to enhance electronic voting, current solutions for blockchain-based electronic voting, and possible research paths on blockchain-based electronic voting systems. Numerous experts believe that blockchain may be a good fit for a decentralized electronic voting system.

Furthermore, all voters and impartial observers may see the voting records kept in these suggested systems. On the other hand, researchers discovered that most publications on blockchain-based electronic voting identified and addressed similar issues. There have been many study gaps in electronic voting that need to be addressed in future studies. Scalability attacks, lack of transparency, reliance on untrustworthy systems, and resistance to compulsion are all potential drawbacks that must be addressed. As further research is required, we are not entirely aware of all the risks connected with the security and scalability of blockchain-based electronic voting systems. Adopting blockchain voting methods may expose users to unforeseen security risks and flaws. Blockchain technologies require a more sophisticated software architecture as well as managerial expertise. The above-mentioned crucial concerns should be addressed in more depth during actual voting procedures, based on experience. As a result, electronic voting systems should initially be implemented in limited pilot areas before being expanded. Many security flaws still exist in the internet and polling machines. Electronic voting over a secure and dependable internet will need substantial security improvements. Despite its appearance as an ideal solution, the blockchain system could not wholly address the voting system’s issues due to these flaws. This research revealed that blockchain systems raised difficulties that needed to be addressed and that there are still many technical challenges. That is why it is crucial to understand that blockchain-based technology is still in its infancy as an electronic voting option.

## Figures and Tables

**Figure 1 sensors-21-05874-f001:**
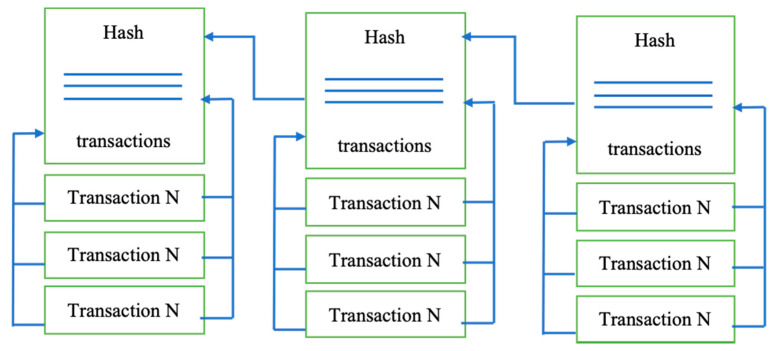
The blockchain structure.

**Figure 2 sensors-21-05874-f002:**
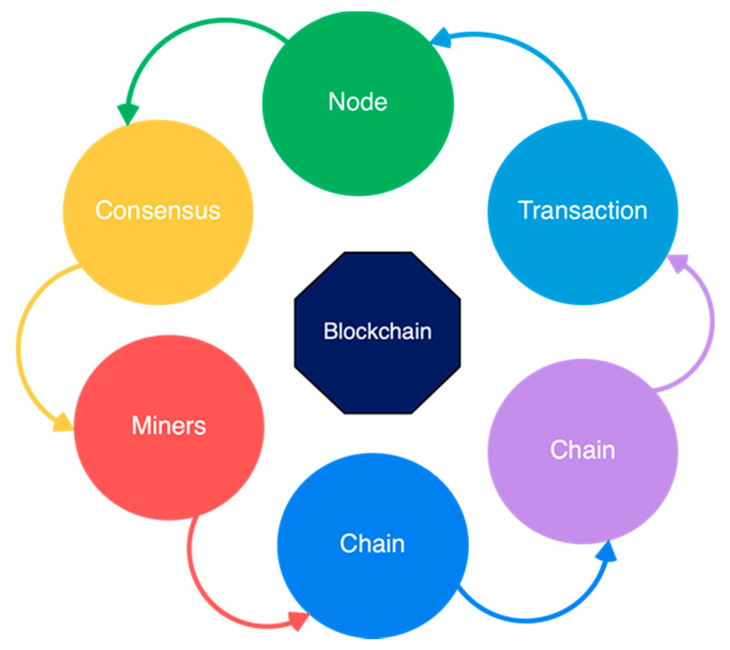
Core components of blockchain architecture.

**Figure 3 sensors-21-05874-f003:**
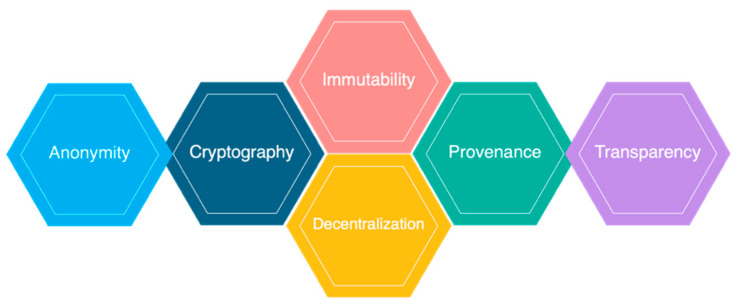
Characteristics of blockchain architecture.

**Figure 4 sensors-21-05874-f004:**
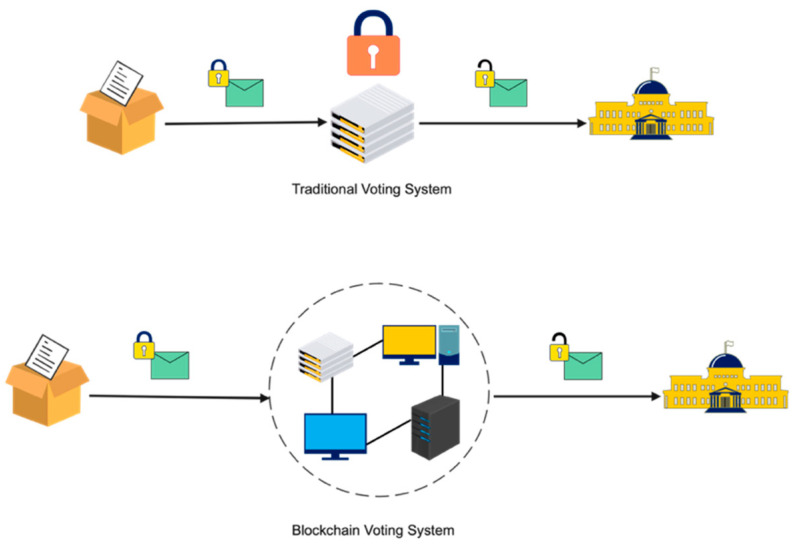
Traditional vs. blockchain voting system.

**Figure 5 sensors-21-05874-f005:**
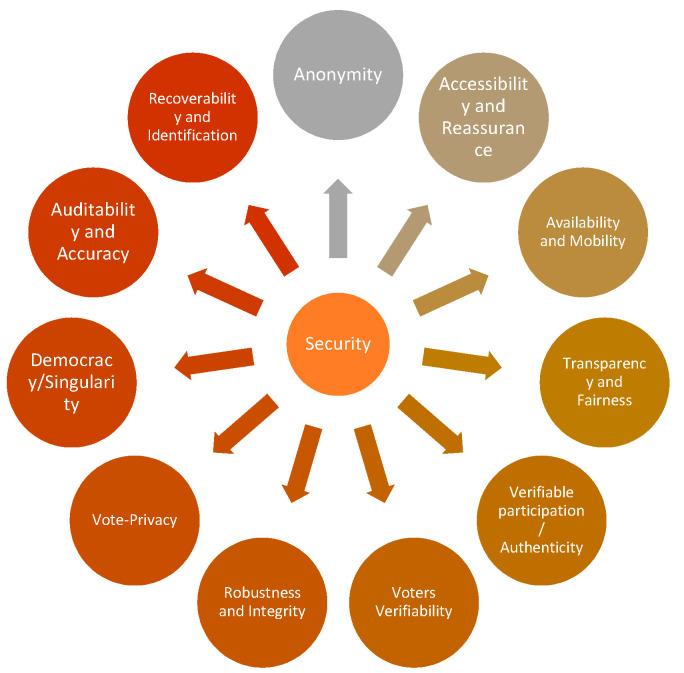
Security requirements for electronic voting system.

**Figure 6 sensors-21-05874-f006:**
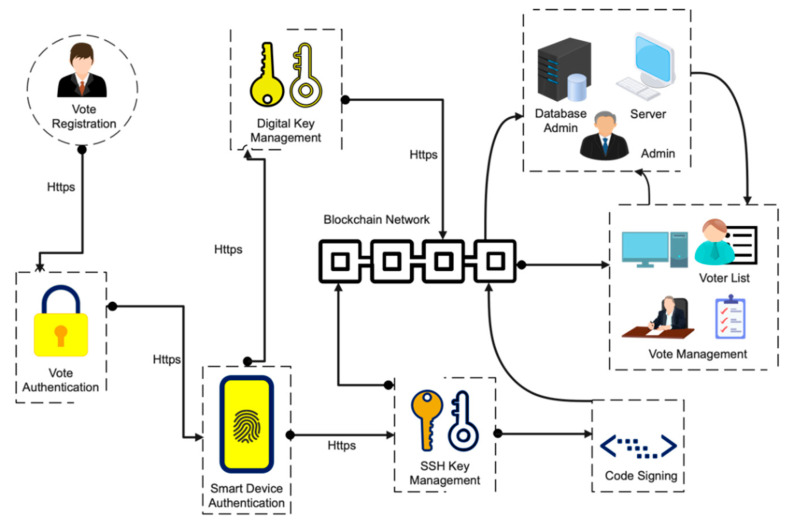
Blockchain voting systems architectural overview.

**Table 1 sensors-21-05874-t001:** Comparison of current blockchain-based electronic voting systems.

Online Voting Platforms	Framework	Language	Cryptographic Algorithm	Consensus Protocol	Main Features (Online Blockchain Voting System)							
					Audit	Anonymity	Verifiability by Voter	Integrity	Accessibility	Scalability	Accuracy/Correctness	Affordability
Follow My Vote	Bitcoin	C++/Python	ECC	PoW	**✓**	**✓**	**✓**	**✓**	**✓**	✘	**✓**	**✓**
Voatz	Hyperledger Fabric	Go/JavaScript	AES/GCM	PBFT	**✓**	**✓**	**✓**	**✓**	**✓**	✘	**✓**	**✓**
Polyas	Private/local Blockchains	NP	ECC	PET	**✓**	**✓**	**✓**	**✓**	**✓**	✘	**✓**	NA
Luxoft	Hyperledger Fabric	Go/JavaScript	ECC/ElGamal	PBFT	**✓**	**✓**	**✓**	**✓**	**✓**	✘	**✓**	**✓**
Polys	Ethereum	Solidity	Shamir’s Secret Sharing	PoW	**✓**	**✓**	**✓**	**✓**	**✓**	✘	**✓**	**✓**
Agora	Bitcoin	Python	ElGamal	BFT-r	**✓**	**✓**	**✓**	**✓**	**✓**	✘	**✓**	**✓**

**Table 2 sensors-21-05874-t002:** Scalability analysis of famous blockchain platforms.

Framework	Year Release	Generation Time	Hash Rate	Transactions Per Sec	Cryptographic Algorithm	Mining Difficulty	Power Consumption	Reward/Block	Scalability
Bitcoin	2008	9.7 min	899.624 Th/s	4.6 max 7	ECDSA	High (around 165,496,835,118)	Very High	25 BTC	Very Low
Ethereum	2015	10 to 19 s	168.59 Th/s	15	ECDSA	High (around 10,382,102)	High	5 ether	Low
Hyperledger Fabric	2015	10 ms	NA	3500	ECC	No mining required	Very Low	No built-in cryptocurrency	Good
Litecoin	2011	2.5 min	1.307 Th/s	56	Scrypt	Low 55,067	Moderate	25 LTC	Moderate
Ripple	2012	3.5 s	NA	1500	RPCA	No mining required	Very Low	Base Fee	Good
Dogecoin	2013	1 min	1.4 Th/s	33	Scrypt	Low 21,462	Low	10,000 Doge	Low
Peercoin	2012	10 min	693.098 Th/s	8	Hybrid	Moderate (476,560,083)	Low	67.12 PPC	Low

**Table 3 sensors-21-05874-t003:** Comparison of selected electronic voting schemes based on blockchain.

Authors	Voting Scheme	BC Type	Consensus Algorithm	Framework	Cryptographic Algorithm	Hashing Algorithm	Counting Method		Security Requirements (Measuring on a Large Scale)						
								Anonymity	Audit	Accuracy/Correctness	Accessibility	Integrity	Scalability	Affordability	Verifiability by Voter
Shahzad and Crowcroft [2]	BSJC	Private	PoW	Bitcoin	Not specified	SHA-256	3rd Party	**✓**	**✓**	✘	**✓**	**✓**	✘	**✓**	✘
Gao, Zheng [8]	Anti-Quantum	Public	PBFT	Bitcoin	Certificateless Traceable Ring Signature, Code-Based, ECC	Double SHA-256	Self-tally	**✓**	**✓**	✘	**✓**	**✓**	✘	**✓**	✘
McCorry, Shahandashti [76]	OVN	Public	2 Round-zero Knowledge Proof	Ethereum	ECC	Not specified	Self-tally	**✓**	✘	✘	**✓**	✘	✘	**✓**	**✓**
Lai, Hsieh [81]	DATE	Public	PoW	Ethereum	Ring Signature, ECC, Diffie-Hellman	SHA-3	Self-tally	**✓**	✘	✘	**✓**	✘	**✓**	**✓**	**✓**
Yi [83]	BES	Public	PoW	Bitcoin	ECC	SHA-256	NA	**✓**	**✓**	✘	**✓**	✘	✘	**✓**	**✓**
Khan, K.M. [86]	BEA	Private/Public	PoW	Multichain	Not specified	Not specified	NA	✘	**✓**	✘	**✓**	✘	**✓**	**✓**	✘

## Data Availability

Not applicable.

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
