# Peer review of "Blockchain for Electronic Voting System—Review and Open Research Challenges"

_sensors, 2021, doi:10.3390/s21175874_

Round 1

Reviewer 1 Report

This is broad review article. The authors invested significant effort to describe state of the art in blockchain based voting systems. However, the article does not have some original contribution from authors. State of the art is correctly elaborated but I do not see that article brings some novel research or at least ideas. I appreciate the effort spent on literature analysis and review but some more scientific ambition is required in order to be published.

Author Response

Thanks for reviewing our manuscript and provide great suggestions. As per the review, We have made all changes in our manuscript outlined below:

  • As per the original contribution: this is a survey paper, and our primary concern by writing this paper was to highlight the pinpoints where the gap is still available to work and research on it. We combined the actual blockchain-based voting systems applications with academic research, analyzed them deeply to compare fields that met the maximum voting requirements, and highlighted the main issues. In section 6, "Electronic Voting on Blockchain" figure: 6, we have provided our model to show how to authenticate the user and store data in a blockchain-based voting system and how the process should work. We are still working on our model and prototype. 
  • Novel Research or at least Ideas: In section 9"Discussion and Future Work," we have highlighted the main areas where more research is needed and categorized them properly where we need to focus in future work. We have provided some solutions for each section for researchers to work on and move on.

We have attached our revised manuscript by following the instruction of other reviewers as well. Please take a look again, and we hope you'll feel it better than before.

Reviewer 2 Report

The paper reviews existing efforts to implement blockchain-based electronic voting systems at scale.

For doing so, it describes the many properties (mostly related to confidentiality, integrity, and security) such a system should have, how the blockchain could provide them, and how to deal with scalability.

The paper is a bit hard to follow at times for 2 main reasons:

  • written English must be improved, and possibly checked by a native speaker, as many sentences are very unusual and not comprehensible (only 2 examples reported below)
    • It is a secured networking technique for unsafe third parties, a theory, and a way to hide information in combination with computer science and mathematics.

    • In short, representational democracy is achievable only with systematize widespread supremacy, and systematize prevalent power means free and fair elections. Electronic voting reveals the entire literature review of the research area of science and similar experiments as one positive direction for making voting a bit simpler both logistically and in terms of citizens'  engagement.

  • organisation can be improved. for instance, I would entirely remove sections 3 and 6, as they do not really add value to the contribution. also, I would swap sections 4 and 5, so as to clearly state what properties to aim for

Besides this, the paper is timely and provides for a much needed contribution, however a major revision is needed to appreciate the survey fully, as follows.

In table 1  I would expect all the properties listed in sections 4 and 5, or at least only those listed in section 5, whereas there are 8 properties partially overlapped with the ones aforementioned but not encompassing all of them.

The same holds for table 2, that shows yet another different set of properties.

This is a big problem for two reasons:

  1. by not comparing the literature (or the commercial platforms) against the properties described before by the same authors in the same paper, those properties lose meaning
  2. by comparing the same thing (blockchain-based electronic voting systems) in two different ways (table 1 vs table 2), that is with respect to different properties, the comparison itself loses meaning

For these reasons I encourage the authors to take into account my comment in their review

Author Response

Thanks for reviewing our manuscript and provide great suggestions. As per the review, We have made all changes in our manuscript outlined below:

All the properties mentioned in this paper are correlated; if we fail to meet the voting requirements and focus on one of them, the rest will be compromised. For these properties, confidentiality, integrity, and security, Blockchain can fulfill partially, but more work is needed because voting is different from other E-govern projects. Our primary focus in this paper is on Blockchain scalability. Because voting is a one-day process, we need a system to handle millions of transactions without interrupting and store all data on multiple locations. As we have mentioned, transaction speed per sec in table 3, "Scalability Analysis of famous Blockchain Platforms." 

  • The paper is hard to follow for two main reasons:  We have read the entire paper, improved the English language, and changed the unusual sentences for better readability.
  • I would entirely remove sections 3 and 6: In section 3, "How Blockchain can Transform the Electronic Voting System," we have further explained in more detail how the Blockchain can transfer and bring changes in current electronic voting systems and showed the difference between both systems in figure 4. In section 6, "Electronic Voting on Blockchain," we have deeply explained how the process should be. In figure 6, we have provided our model to show how to authenticate the user and store data in a blockchain-based voting system and how the process should work. Moreover, we are working on our prototype to implement this model for making a Blockchain-based voting system more secure, scalable, and easy, so to support the novelty of our work, we would like to keep the section in this paper.
  • In table 1, Properties: In section 4, these are not security requirements for voting. These are conditions that need to be satisfied for developing electronic voting systems. However, for section 5, yes, correct. We have explained 11 security requirements for a voting system. These are general security requirements for every election rather than the paper base, EVM (electronic voting machine), electronic or blockchain-based electronic voting systems. We have merged some of them because they are correlated to each other and not require for blockchain-based electronic voting systems, for example: 
  • Robustness and Integrity, Vote Privacy -> Integrity
  • Auditability and Accuracy, Lack of Evidence, Transparency and Fairness, Recoverability and Identification -> Immutability

That is why we merge security requirements and conditions to analyze some of the current available blockchain-based electronic voting systems.

  • The same holds for table 2: Yes, correct as we explained earlier, all security requirements are not feasible on electronic or blockchain-based electronic voting systems. We merge conditions with requirements because most of the requirements are already fulfilled by Internet/electronic and popup some new challenges and issues.

However, we accept your highlighted issues about Tables 1 and 2 about using different words for the same properties. We have reviewed both tables and match the property names and make them correct.

Reviewer 3 Report

I have thoroughly reviewed the paper and have no further comments.
I only suggest to improve the clarity of the figures in the paper. 

Author Response

Thanks for reviewing our manuscript and provide great suggestions. As per the review, We have made all changes in our manuscript outlined below:

Clarity of the Figures:

  • As per suggestion, We have added some content to explain the figures.

Reviewer 4 Report

This paper provides a review about the technologies for blockchain-based electronic voting and identifies some open issues in this area for future research development.

Overall, the paper is well organized, but some polishing on English writing is needed for further enhancing the readability. Also, the discussions about possible future research directions given in Section 8 are fairly brief; therefore, more elaboration on these open issues and possible technical solutions would make this paper more beneficial to the readers. 

Author Response

Thanks for reviewing our manuscript and provide great suggestions. As per the review, We have made all changes in our manuscript outlined below:

English changes:

  • We have made some changes for better readability in English to understand the core concept of this paper.
  • Section 8: We have added more content to elaborate on open issues and provide some possible technical solutions to solve these issues.

Reviewer 5 Report

The paper is a survey and analysis on the application of blockchain technology for electronic voting systems.

The authors point out the difficulties related to the implementation of the Blockchain system in the case of electronic voting and identify the requirements that current systems meet satisfactorily, but also the requirements for which there are no valid solutions yet.

As a general remark, the paper is well organized and structured. However, the reviewer would like to make some recommendations.

In general, English is correct and clean. However, authors should review certain portions of text that, in the reviewer’s opinion, require rephrasing. Few examples of such phrases follow:

  • Page 2, lines 64-65: “Where technology brought the Electronic Voting method [9], which is very important and has caused major difficulties in the democratic system.”
  • Page 2, lines 69-71: “Nonetheless, the current electronic voting protocols risk over-authority and manipulated details, preventing true fairness, privacy, secrecy, anonymity, and openness in the voting process.”
  • Page 9, lines 390-391: “Moreover, the basic concepts of building the solution, but in a real-world system many more factors need to be taken into account: …”
  • Page 11, lines 443-444: “The voter who voted they want to reassurance that their votes will not be tampered with or revealed.”

Technical issues:

  • Page 2, end of section1: even if the paper is a survey, the authors should highlight at the end of the introductory section the original contributions of the paper, in the context of the topic addressed.
  • Page 6: the simple reference to “the below Figure 4” is insufficient. The authors must add text to explain the mechanisms in Figure 4.
  • Page 13, lines 506-511: pay attention to repeated text. Text from lines 499-504 is repeated in lines 506-511.
  • Page 14, Table 1: the authors are invited to comment on the unavailability of the “Scalability” feature for all online voting platforms presented in Table 1.
  • Page 16, Table 2: the authors are invited to comment on the unavailability of most security requirements for most schemes in Table 2.

Author Response

Thanks for reviewing our manuscript and provide great suggestions. As per the review, We have made all changes in our manuscript outlined below:

General Issues:

  • Page 2, lines 64-65: Completely modified
  • Page 2, lines 69-71: Rephrased
  • Page 9, lines 390-391: Rephrased
  • Page 11, lines 443-444: Rephrased

Technical issues:

  • Page 2, end of section1: Added new section
  • Page 6, added a new paragraph to explain figure 4
  • Page 13, lines 506-511: Removed duplicate content.
  • Page 14, Table 1: Added content to explain why the currently available platforms are not scalable to handle national-level elections.
  • Page 16, Table 2: Added a new property to match with table 1

Round 2

Reviewer 1 Report

Authors made significant effort and presented extensive literature review about blockchain technology and electronic voting systems. The paper can be published.

Author Response

Thank you so much to appreciate my work.

Reviewer 2 Report

A part from the consideration that I still think sections 3 and 6 do not add much, the authors have addressed most of my major concerns, and I wont'oppose publication anymore

Author Response

Thank you so much for appreciate our work.